# Significance of Machine Learning for Detection of Malicious Websites on an Unbalanced Dataset

**Ietezaz Ul Hassan [1], Raja Hashim Ali [1,\*], Zain Ul Abideen [1], Talha Ali Khan [2,\*] and Rand Kouatly [2]**

[1] Faculty of Computer Science and Engineering, Ghulam Ishaq Khan Institute of Engineering Sciences and Technology, Topi 23460, Pakistan

[2] Faculty of Tech and Software Engineering, University of Europe of Applied Sciences, 14469 Potsdam, Germany

\* Correspondence: hashim.ali@giki.edu.pk (R.H.A.); talhaali.khan@ue-germany.de (T.A.K.)

**Abstract:** It is hard to trust any data entry on online websites as some websites may be malicious, and gather data for illegal or unintended use. For example, bank login and credit card information can be misused for financial theft. To make users aware of the digital safety of websites, we have tried to identify and learn the pattern on a dataset consisting of features of malicious and benign websites. We treated the problem of differentiation between malicious and benign websites as a classification problem and applied several machine learning techniques, for example, random forest, decision tree, logistic regression, and support vector machines to this data. Several evaluation metrics such as accuracy, precision, recall, F1 score, and false positive rate, were used to evaluate the performance of each classification technique. Since the dataset was imbalanced, the machine learning models developed a bias during training toward a specific class of websites. Multiple data balancing techniques, for example, undersampling, oversampling, and SMOTE, were applied for balancing the dataset and removing the bias. Our experiments showed that after balancing the data, the random forest algorithm using the oversampling technique showed the best results in all evaluation metrics for the benign and malicious website feature dataset.

**Keywords:** machine learning; malicious website; benign website detection

## 1. Introduction

Digital security has gained paramount importance in recent times with the exponential growth in the number of applications and users, and rapid evolution in the field of Information Technology. Easy access to the internet from across the globe, the availability of high-speed Internet, and technological advances through the availability of 4G and 5G technology, have significantly increased usage of the Internet around the world [1]. Specifically, due to the recent waves of COVID-19 pandemic, several companies and businesses shifted their business models from the physical domain to the digital domain, using web applications and mobile applications to reduce physical contact [2,3]. However, with the opportunity to grow significantly and be open and accessible to the world, there is a significant security threat as well—the leaking of private or insecure data. Some recent major data leaks involve large volumes and variety of compromised data and have impacted millions of online users (as summarized in Table 1).

The openness and ease of access to the Internet has significantly increased the digital visibility of a person, which leads to opportunities for hackers and digital thieves to gain access to private credentials and data. This is a severe breach of security, leading to financial loss and deep mental pressure at times. One way to gather private data from unsuspecting internet users is through malicious websites. The malicious websites typically look like ordinary benign websites and ask for private data, for example, credit card information, or usernames and passwords to gain access to private pictures, or other important information. This information is kept stored in a database and can then be used online for any malicious

purpose, for example, for online shopping, the illegal transfer of money, or for blackmailing or harassing the person. One of the simplest techniques to steal digital information is using identical-looking fake pages to the original web pages [4].

**Table 1.** Table displaying the recent large data breach attacks, their impact, and their cause. Digital security, especially for online users and businesses, is a significant cause of concern.

| Breach Name | Date of Data Breach | Impact | Caused by |
| --- | --- | --- | --- |
| Ronin (Ethereum sidechain to power Axie Infinity) Breach | March 2022 | Looted over 540 million USD | Hackers (Lazarus Group, North Korea) |
| 0ktapus | August 2022 | Compromised at least 130 companies (including Cloudflare, Doordash, Mailchimp) | Extended Phishing Campaign |
| Uber Total Compromise | August 2022 | Complete access to Uber's source code, internal databases, and more information by a hacker under the alias "teapotuberhacker" | Hacker with ties to Lapsus$ using purchased credentials and MFA fatigue attack |
| Lapsus$ hacking spree | February–March 2022 | Looted a terabyte of proprietary data (Nvidia) and blackmail the company. Leaked source codes and algorithms from Samsung. Temporarily brought down Ubisoft's online gaming services. Partial source code released for Bing and Cortana, breaching Microsoft Inc. | Method not known |
| Neopets Breach | 19 July 2022 | Personal data of 69 million Neopets users including username, email addresses, date of birth, zip codes was released | Phishing attack |

One way to ensure the security of the user is to identify if the accessed website is malicious or not, using classification techniques [5,6]. An accurate classification ensures that the user will be warned not to enter data on the suspicious website [7,8]. Machine learning techniques have recently shown excellent results when used for the classification of data [9,10]. They are not only limited to the field of malicious attacks but have been used for many prediction and identification tasks in multiple fields including image processing [11], weather prediction [12], price prediction [13], stock prediction [14], and other topics. With the availability of large amounts of data, improvement in computing power, and the development of advanced models of computation, the field of machine learning has shown a lot of promise and progress. In traditional machine learning approaches, a new model is developed first and its architecture and parameters are initialized. It is then trained using the training dataset so that the model learns the mapping between the input features and the expected output. After learning the intrinsic general mapping between the input feature set and the output labels, the model can then be used for the classification of the unseen dataset with a similar feature set. This phase is called the validation phase, where the model is validated during training by measuring its performance on unseen data. A trained model can then be applied to unseen data and the results are measured to evaluate the performance. Some of the popular machine learning algorithms are K-Nearest Neighbors (KNN) [15], Support Vector Machine (SVM) [16], Decision Tree [17], Logistic Regression [18], and Naïve Bayes [19].

Several evaluation metrics, e.g., accuracy, precision, and recall, are used in the literature to measure the performance of a machine learning model [20]. All these metrics are typically derived from the confusion matrix, where the idea is to count true positives, true negatives, false positives, and false negatives. Then, a specific formula based on these four counts and their ratio can be used to evaluate the performance of an algorithm on a given dataset. Since the false positives or false negatives are both measures meant for counting the

incorrect classification of data, while the true positives or the true negatives are measures for counting the correct classification of data, all ratios generally tend to improve the true positives or the true negatives, or both, while lowering the count of false positives or false negatives or both.

However, one of the major issues faced during the classification of data is the imbalanced dataset [21,22]. In an unbalanced dataset, a single class or a selected group of classes contained the most samples and dominated the data. This means that if a method gets biased toward a certain class or group of classes with the most data, then it will give good results for that class and simply ignore other classes. Several strategies for data balancing are proposed in the literature. SMOTE, undersampling, and oversampling are some of the popular methods that have been deployed for data balancing [23].

In this study, machine learning techniques were deployed for the identification of malicious and benign websites. We used the "Malicious Website" data set that is publicly available on Kaggle. The dataset consists of features of websites that can be used to determine if the website is malicious or benign. We trained five different machine learning models on this dataset. The goal of machine learning models is to capture the underlying structure of the data. When the underlying structure of the data has been captured and a new unseen record is presented to the model, the machine learning model can determine whether the new set of features should be labelled malicious or benign.

Note that the "Malicious Website" dataset is imbalanced; when the model is trained on such data, it is biased towards the class whose records are in a majority because it is rewarded to classify all data as members of that class. So, for resolving the issue of data imbalance and the skew toward malicious websites, data balancing techniques including undersampling, oversampling, and SMOTE were used in this study to improve the performance of the model [21].

As discussed earlier, the machine learning model had two phases, a training and validation phase, followed by a test phase, where the dataset is divided into training data for training and validating the model and test data for measuring the performance of the model. The model is trained on training data and then its performance is tested on the test data. Note that fixing the datasets into a fixed test and train dataset sometimes causes issues for a particular division of the data, for example, when all classes are not evenly distributed among the two subsets. Hence, k-fold cross-validation is generally recommended to counter these complications with continuous changing of validation and training data samples in each iteration of training.

A 10-fold cross-validation of the dataset for training and validation was deployed in this study. We evaluated the model performance based on the five most common metrics including accuracy, precision, recall, F1-score, and true positive rate. After checking the performance of classifiers on the dataset, the study recommends that deploying the random forest technique when used with oversampling for balancing the dataset gave the best results for all metrics.

## 2. Related Work

Singhal et al. [24] used several supervised machine learning classifiers, such as random forest, gradient boosting, decision trees, and deep neural networks, for the classification of malicious and benign websites. First, URLs were collected. From each of the malicious and benign websites, the authors extract lexical-based, host-based, and content-based features for the website, which served as input for the machine learning models. The lexical-based features selected by the authors are URL length, host length, host token count, path length, and several symbols. Similarly, the host-based features extracted from the URL are location and autonomous system number (ASN). The content-based features selected by the author are HTTPS-enabled, applet count, Eval() function, XMLHttpRequest (XHR), popups, redirection, and unescaped() function. The authors collected the benign website from the public blacklist provided by PhishTank. There are a total of eighty thousand unique URLs in this dataset and the dataset is balanced. After collecting the data, the

features are extracted. The evaluation metrics used for measuring classifier performance on this dataset are accuracy, precision, and recall for comparing various classifiers. The paper achieved the best result of 96.4% accuracy by using the gradient boosting technique.

Patil et al. [25] designed their algorithm, called kAYO, for distinguishing between malicious and benign mobile webpages. Their method uses the static features of a webpage for classification. The authors also applied their method to a large, labelled dataset, made up of 350,000 malicious and benign mobile webpages, on which the authors achieved an accuracy of 90 per cent. The authors also developed their browser extension. At the backend of the browser extension, kAYO is running for identifying whether the selected webpage is malicious or benign.

Iv et al. [26] explored the relationship between the number of extracted features from the HTTP header and the chance of detecting malicious websites. They analyzed HTTP headers of 6021 malicious and 39,853 benign websites. From these websites, the authors extracted 672 features and identified 22 features for further analysis, of which 11 features were studied in prior research while the remaining 11 features were identified in their work. Of these 22 features, three features accounted for 80% of the total importance of all the features. The authors observed that instead of using only 11 features as was performed initially, a better result is observed if all 22 features are used. Furthermore, the authors also applied two dimensionality reduction techniques, in which it was observed that the application of principal component analysis (PCA) on the identified features increases the detection. The authors used eight supervised machine learning classifiers in this work.

Patil et al. [27] used a hybrid methodology for the detection of malicious URLs. The hybrid methodology stands for a combination of static and dynamic approaches, in which some features were extracted using a static approach and some were extracted using a dynamic approach. The authors extracted a total of 117 features, of which 44 were new features. The dataset used in this paper consisted of 52,082 samples. The training data consisted of 40,082 samples out of which 20,041 were malicious and 20,041 were benign. This shows that the dataset used by the authors in this study was balanced. For the effective detection of malicious website URLs, the authors built their classifier using a majority voting classification scheme. The authors evaluated their method using six decision tree classifiers including the J48 decision tree, Simple CART, random forest, random tree, ADTree, and REPTree. The authors used accuracy, false-positive rate, and false-negative rate for evaluation. By using their majority-voting classification method, the authors were able to achieve an accuracy of 99.29% with a low false positive rate and a low false negative rate. The authors showed that with decision tree-based classification, the authors achieved an accuracy between 98 to 99 per cent. The authors have also compared their results with 18 anti-virus and anti-malware solutions, and show promising results for their proposed methodology.

Al-milli et al. [28] proposed a one-dimensional convolutional neural network (1D-CNN) model for the identification of illegitimate URLs. The authors used benchmark datasets and two evaluation metrics (accuracy and receiver operating characteristic ROC curve) for their experiment. Their proposed model achieved an accuracy of 94.31% and an area under curve (AUC) value of 91.23%. Sixty-four filters, having a kernel size of 16 each, were applied in the proposed CNN architecture. The authors used the rectified linear unit (ReLU) activation function that was followed by a max-pooling layer. The final layer was fully connected where there was only one neuron and used sigmoid as the activation function. In their dataset, there were a total of 2456 records having 30 features. In the dataset, there were three output classes which were false URL, true URL, and suspicious. The authors also considered the suspicious and false URLs in the same category. They used 70 per cent of the dataset for training and 30 per cent of the dataset for testing. They used 500 and 2000 epochs. By increasing the epochs, their results were improved by a rate of 11.31 per cent.

Jayakanthan et al. [29] proposed a method for the detection of malicious URLs in two steps. The first step is the enhanced probing classification (EPCMU) algorithm to detect

a malicious URL. The second step is the naïve bayes. Detection is performed in the first step and classification is performed in the second step. The EPCMU checks the input URL with very deep details. If any feature of a malicious website is found, or it is found in the list of the blacklisted profile of the system, it then reports this URL as malicious. Otherwise, it checks in depth further. In the classification step, the naïve bayes algorithm takes input from the EMPCU, which is a set of URLs. It checks whether the URL set is malicious or genuine.

Assefa et al. [30] proposed an auto-encoder for differentiating between a malicious and benign website. The data for phishing websites was collected from Phish Tank, an open-source dataset, and the data for genuine websites was collected from the Canadian Institute for Cybersecurity dataset. The final dataset consisted of 10,000 samples with 16 features in total. Features are initially extracted from the data and it is then preprocessed to remove incomplete data. Autoencoders are trained only on URLs of legitimate websites so that when an unseen validation URL is encountered, it will be classified based on the amount of deviation from the typical characteristics of a benign website. The autoencoder was made up of three layers namely input, hidden, and output layers. The authors compared the performance of their model with the SVM and decision trees. Their model, based on autoencoders, gives an accuracy of 91.24%. While the SVM and decision tree algorithms give an accuracy of 88.4% and 86.1%, respectively, the accuracy of both traditional machine learning algorithms is significantly lower than that of the autoencoder.

Table 2 displays the literature review in terms of the different research works that have worked on separating benign websites from malicious websites.

**Table 2.** Table displaying the literature review for important research works in the field of malicious and benign website identification and their contribution to the field.

| Author Name | Balanced Dataset | Model | Metric | Metric Value (Accuracy) |
|---|---|---|---|---|
| Singhal et al. [24] | Yes | Random forest, Gradient boosting, Decision trees, Deep neural networks. | Accuracy, Precision, Recall. | 96.4% |
| Patil et al. [25] | | kAYO (self-Proposed) | Accuracy. | 90 % |
| Iv et al. [26] | No (SMOTE) | Adaptive Boosting, Extra Trees, Random Forest, Gradient Boosting, Bagging Classifier, Logistic Regression, K-Nearest Neighbors. | Accuracy, False positive rate, False negative rate, AUC. | 89% |
| Patil et al. [27] | Yes | J48 decision tree, Simple CART, Random forest, Random tree, ADTree, REPTree. | Accuracy, False-positive rate, False-negative rate. | 99.29% |
| Al-milli et al. [28] | | One-dimensional Convolutional Neural Network (1D-CNN) | Accuracy, ROC curve. | 94.31% |
| Jayakanthan et al. [29] | | Enhanced Probing Classification (EPCMU) for detection, Naïve Bayes. | | |
| Assefa et al. [30] | | Auto-encoder, SVM, Decision trees. | Accuracy. | 91.24% |
| Vinayakumar et al. [31] | | Deep Learning. | Accuracy. | 99.96% |
| Vazhayil et al. [32] | | CNN-LSTM. | Accuracy. | 98% |

## 3. Methodology

In this study, we used the publicly available "Malicious Websites" dataset on the Kaggle website. The original dataset is imbalanced with a strong bias towards malicious websites. For solving the imbalanced dataset problem, we applied the three data balancing techniques—undersampling, oversampling and SMOTE. After making the dataset balanced, the K Fold cross-validation technique was then applied for evaluating the performance of the model. In this paper, we used five machine learning classifiers including decision trees, random forest, the support vector machine (SVM), logistic regression, and stochastic gradient descent. The complete methodology and workflow of our contribution are discussed in Figure 1.

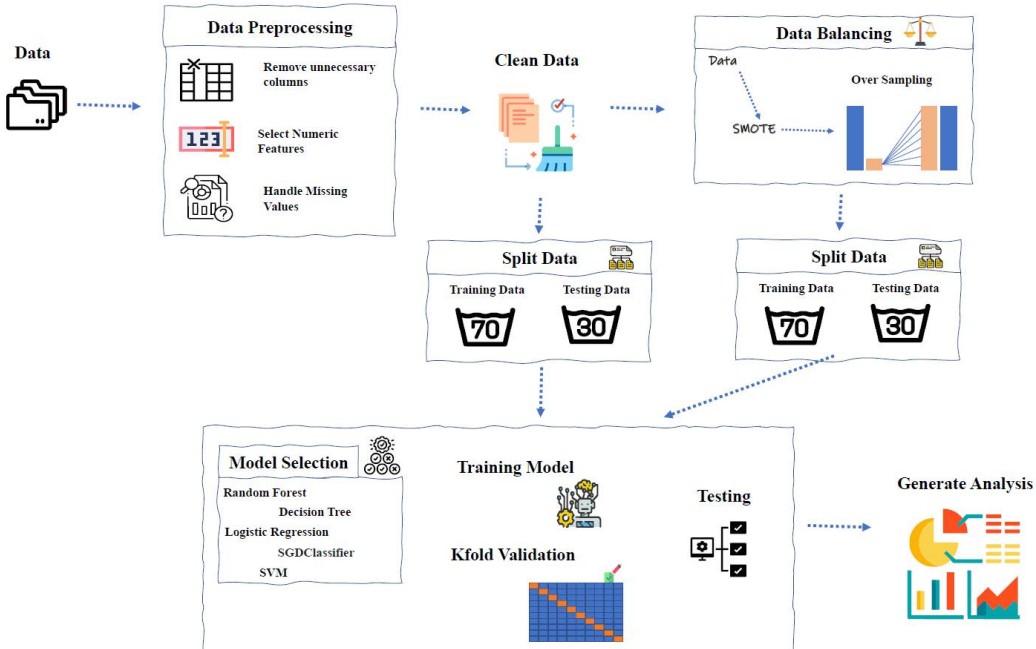

**Figure 1.** The overall working of the proposed solution.

### 3.1. Dataset Description

The dataset used in this study consists of 1781 records of malicious and benign website data with 13 features (independent variables), while the target label column 'Type' indicates whether the sample website is malicious or not. Features used for predicting whether the website is malicious or benign are 'length of URL', 'special character number', 'content-length', 'TCP conversation exchange', 'destination remote TCP port', 'remote IPS', 'APP bytes', 'source app packets', 'remote app packets', 'source app bytes', 'remote app bytes', 'App packets', and 'DNS query time'. The different characteristic features of the dataset are shown in Figure 2.

### 3.2. Data Balancing Technique

In the "Malicious Website" dataset, there are a total of 1781 samples, of which there are 1565 samples that correspond to the malicious class and the remaining 216 samples belong to the benign class. The pie chart in Figure 3 displays the data distribution based on whether the sample belongs to the malicious website class or the benign website class.

From Figure 3, it can be observed that the data is imbalanced and heavily biased toward malicious websites. So, for addressing the imbalanced dataset, three data balancing techniques, namely undersampling, oversampling and the SMOTE, are used to handle the imbalance in the data. Each of these techniques are discussed below.

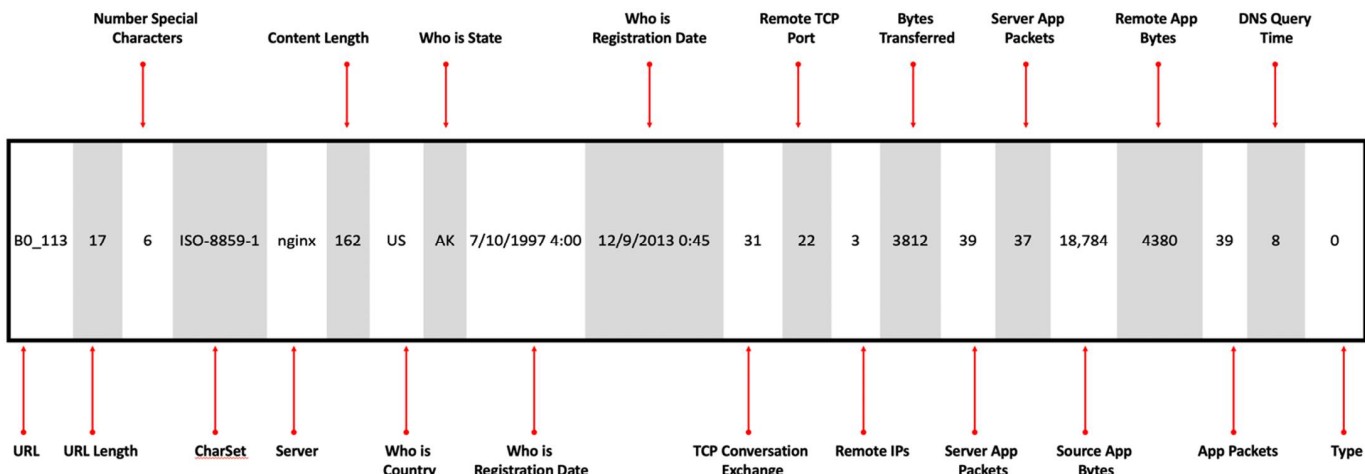

**Figure 2.** Figure showing a sample row of the data and their column labels.

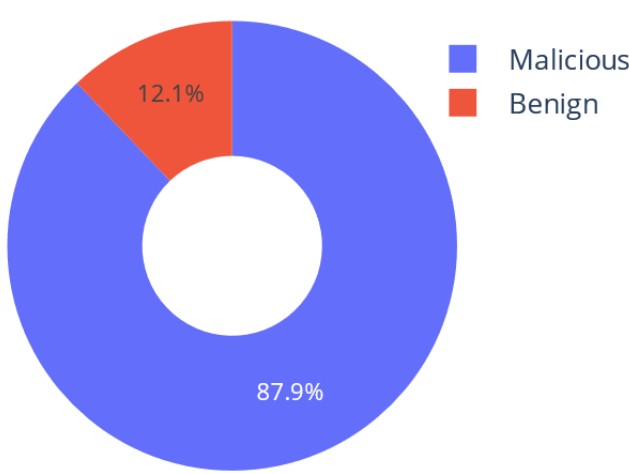

Percentage of Malicious and Benign Websites

**Figure 3.** Pie chart showing the distribution of malicious and benign website samples in the "Malicious Websites" Kaggle dataset.

### 3.2.1. Random Undersampling Technique

In undersampling, the dataset is balanced by reducing the size of the majority class to make it equal to the minority class. As the minority class in this dataset contained 216 records, the majority class (samples corresponding to the malicious data) were reduced to 216 from 1565 by randomly choosing values. The new dataset consists of 432 samples only, of which 216 samples are labelled as malicious and the remaining 216 samples belong to the benign class.

### 3.2.2. Random Oversampling Technique

In oversampling, the dataset is balanced by duplicating the samples of the minority class, so that it becomes equal to the number of samples in the majority class. Therefore, the number of samples in the benign class was duplicated so that the 216 benign class samples became 1565. The samples to be duplicated were randomly selected from the original 216 samples, and a duplicate for each selected sample was added to the dataset. After the oversampling operation, the dataset contains a total of 3130 samples with 1565 samples labelled as malicious websites and 1565 samples belonging to the benign class.

### 3.2.3. Synthetic Minority Oversampling Technique (SMOTE)

The synthetic minority oversampling technique (SMOTE) is an oversampling technique that begins by randomly selecting a minority class instance and locating its k-nearest minority class neighbors. The synthetic instance is then constructed by selecting one of the k nearest neighbors b at random and connecting a and b in the feature space to form a line segment. The synthetic instances are created by convexly combining the two selected examples a and b. After applying the SMOTE, our dataset contains 3128 records, of which 1564 records belong to the malicious class and 1564 records belong to the benign class.

### 3.3. Classifiers

Since this work is based on a machine learning-based classification mechanism, several classifiers were tested to identify the best-performing classifier. The following sections give a brief introduction to the various classification techniques commonly used in the field of classification.

### 3.3.1. Decision Trees

Decision trees belong to the family of supervised learning algorithms. Unlike other supervised learning algorithms, the decision tree algorithm can also be used to solve regression and classification problems [33]. The goal of using a decision tree is to build a training model that can predict the class or value of a target variable by learning simple decision rules from prior data (training data). To predict a class label for a record in decision trees, we start at the root of the tree. We compare the values of the root attribute and the record attribute. Based on the comparison, we proceed to the next node by following the branch corresponding to that value. Decision trees classify examples by descending the tree from the root to some leaf/terminal node, with the classification provided by the leaf/terminal node. Each node in the tree represents a test case for some attribute, and each edge descending from the node represents one of the possible answers to the test case. This recursive process is repeated for each new node-rooted subtree.

### 3.3.2. Random Forest

Random forest is a popular supervised machine learning algorithm for classification and regression problems. It builds decision trees from various samples and classifies them based on their majority vote. The random forest algorithm's ability to handle data sets with both continuous and categorical variables, as in regression and classification, is one of its most important features. In classification problems, it outperforms other algorithms [34]. The following are some of the steps involved in the random forest:

- Random forest selects $n$ random records at random from a data set of $k$ records.
- A distinct decision tree is constructed for each sample.
- Each decision tree yields a result.
- In classification, the final result is determined by majority voting.

Figure 4 illustrates the working of the random forest algorithm on the test dataset.

### 3.3.3. Logistic Regression

Logistic regression, a probabilistic statistical method, is a popular supervised machine learning algorithm used for classification and optimization problems [35]. The algorithm has shown great performance for a variety of common applications such as email spam detection, diabetes prediction, cancer detection, etc. In logistic regression, the sigmoid function (also called the logistic function) and a threshold are used to calculate the likelihood of a label.

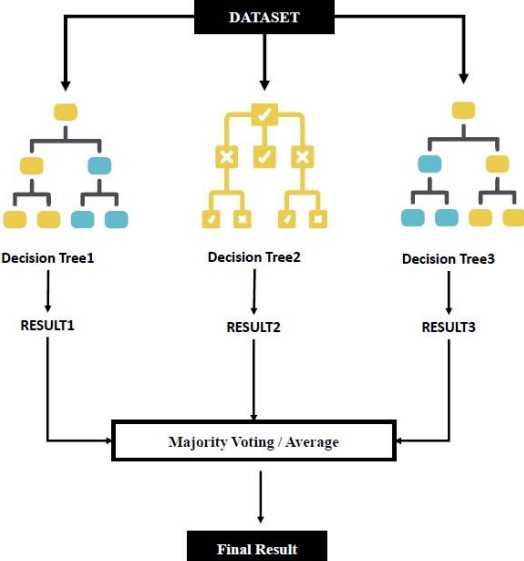

**Figure 4.** An illustration of how random forest works as a classification technique.

Logistic regression differs from linear regression in multiple ways. Linear regression assumes a linear relationship between the dependent and independent variables. The best fit line describes two or more variables in linear regression. Moreover, linear regression attempts to predict the outcome of a continuous dependent variable with high accuracy. On the other hand, logistic regression predicts the likelihood of an event or class that is dependent on other factors. Logistic regression estimates the likelihood of each label for the test sample and is typically deployed for predicting the target value with categorical dependent variables, for example, with binary labels ('true' or 'false', 'yes' or 'no'). Since the prediction of logistic regression is a likelihood value, it forms an "S" shape when plotted on a graph due to likelihood ranged between 0 and 1. The working of the logistic regression model is shown schematically in Figure 5.

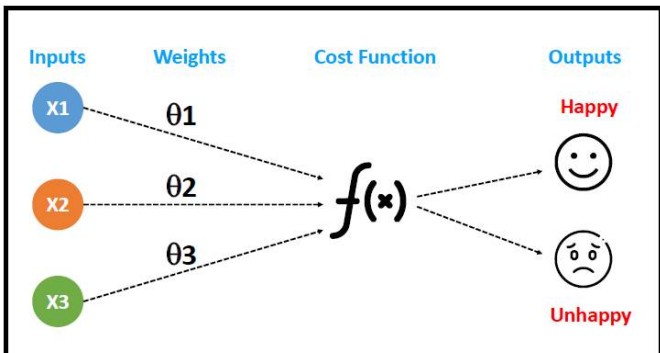

**Figure 5.** An illustration of how the logistic regression algorithm works as a classification technique.

### 3.3.4. Support Vector Machine

The support vector machine algorithm's goal is to find a hyperplane in an N-dimensional space. To separate the two types of data points, a variety of hyperplanes could be used. The SVM looks for the plane with the smallest margin of error. The margin is the difference in distance between two groups of data points. Increasing the margin distance provides some reinforcement, making future data points more confidently classified. Hyperplanes are decision boundaries that aid in the classification of data points. Data points on either side of the hyperplane can be classified in a variety of ways. Furthermore, the number of features determines the hyperplane's size. When only two input features are present, the hyperplane is simply a line. When the number of input features reaches three, the

hyperplane transforms into a two-dimensional plane. When the number of features exceeds three, it becomes difficult to imagine. Support vectors are data points that are closer to the hyperplane and have an effect on its position and orientation. Using these support vectors, we maximize the classifier's margin. The position of the hyperplane will change if the support vectors are removed. These are the considerations that will aid in the development of the SVM model [36]. The working of the SVM is shown schematically in Figure 6.

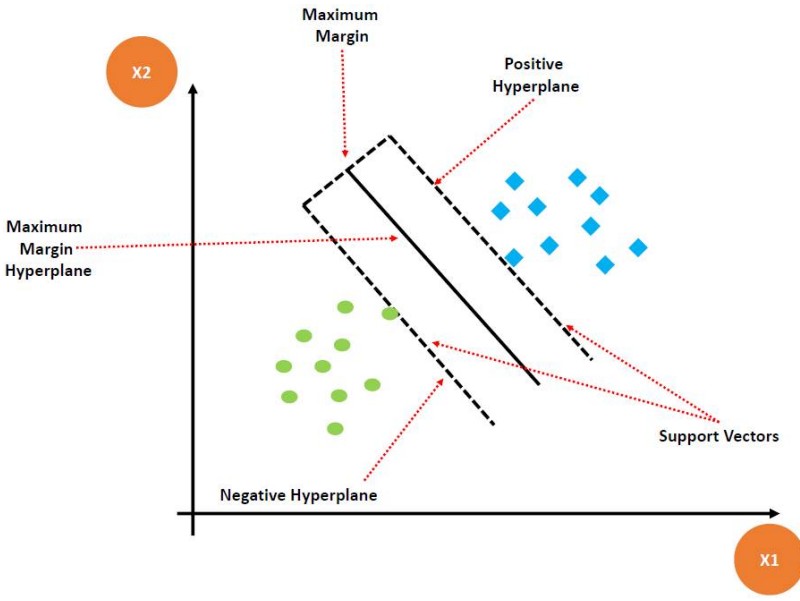

**Figure 6.** An illustration of how support vector machine (SVM) works as a classification technique.

3.3.5. Stochastic Gradient Descent

Gradient descent is a popular algorithm used in artificial neural networks to back-propagate errors during the training of neural networks. It is one of the most commonly used algorithms that minimizes error functions. The gradient descent starts an iterative process with an initial set of parameters and iteratively moves towards a set of parameter values that tend to find the local minima of an error function. Gradient descent is based on the derivatives of gradients that can help reach the global minimum. However, note that the gradient descent algorithm is extremely slow on very large networks and can lead to vanishing gradient problems for large networks. For each iteration of the gradient descent algorithm, a prediction of each instance in the training dataset is required. The procedure could take a long time when dealing with millions of samples and a billion data points per sample. Stochastic gradient descent differs significantly from gradient descent because the coefficient update for the algorithm occurs only during the execution of the training process. Note that the update procedure for the coefficient remains the same as that of the gradient descent algorithm, except for the custom, which is summed for one training sample instead of overall samples. This is the main difference between gradient descent and stochastic gradient descent for classification [37]. The working of the stochastic gradient descent algorithm in searching the solution space and converging to a solution is shown schematically in Figure 7.

*3.4. K-Fold Cross-Validation*

The cross-validation approach is a resampling strategy for testing machine learning models on a small dataset and estimating their efficacy. The technique of cross-validation is used to determine the accuracy of an untested machine learning model, i.e., the test and training data are continuously swapped in each iteration. The concept of swapping the validation data continuously helps in evaluating how well the model learns the general characteristics of the data. The procedure takes one input, k, which specifies how many

subsets of the original data set are to be created. Accordingly, the process is sometimes referred to as k-fold cross-validation. For example, k = 10 denotes a "10-fold cross-validation," where k is the number of folds. The general mechanism of k-fold cross-validation is shown schematically in Figure 8.

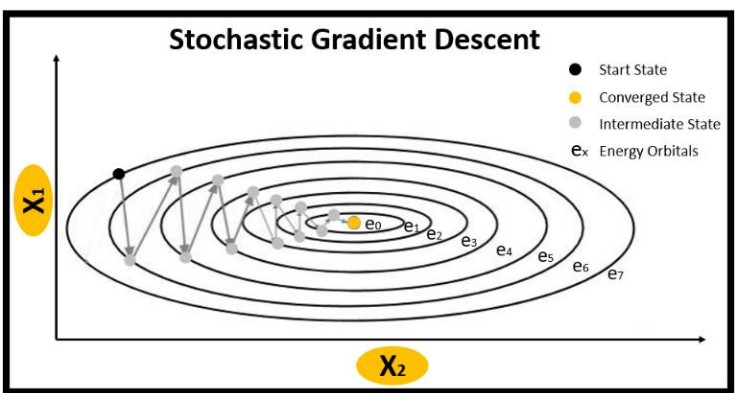

**Figure 7.** An illustration of how stochastic gradient descent works as an optimization strategy for reaching global optima.

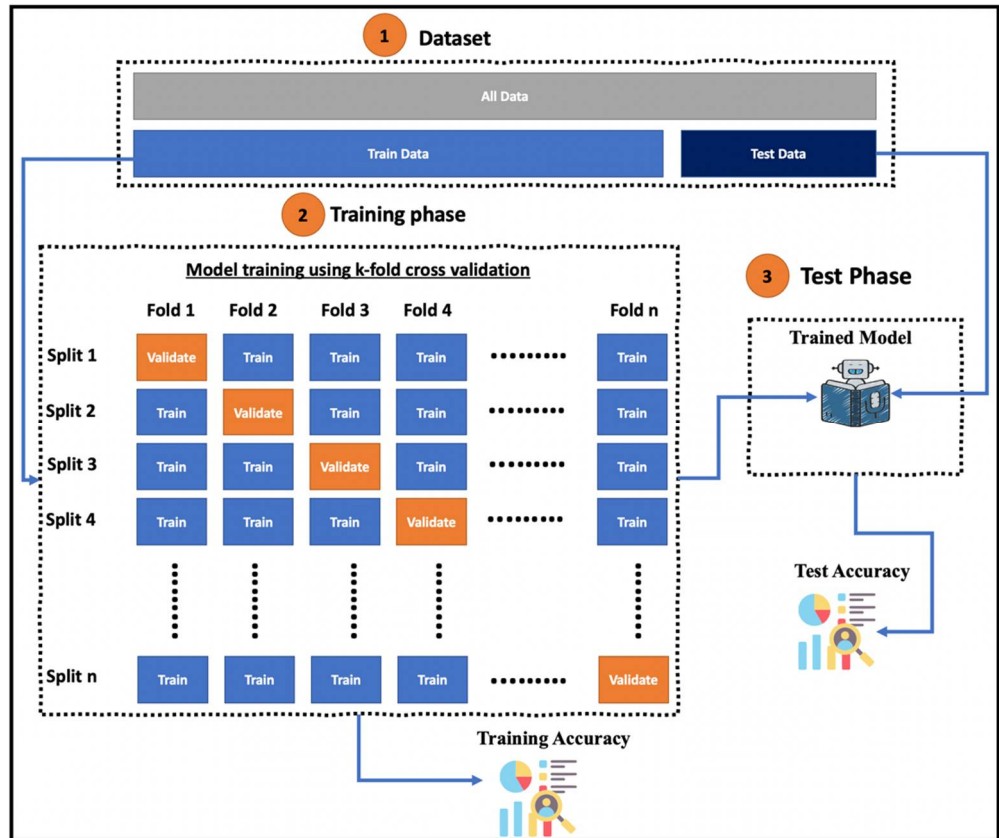

**Figure 8.** An illustration of how the data is divided into *n* folds during k-fold cross-validation.

### 3.5. Overall Methodology

In this study, we tested five different classifiers with no data balancing, the SMOTE, oversampling, and undersampling strategies for removing bias from each class, if any. We deployed a 70/30 split for training and test data. From the training data, we used a stratified 10-fold validation scheme. Nine folds from the training data were used to train the classifier, while the tenth fold of the training data was used to validate the training progress. After the training was complete, the results were then verified on the processed

and balanced test data. The overall process of 10-fold cross validation technique deployed for training is shown in Figure 9.

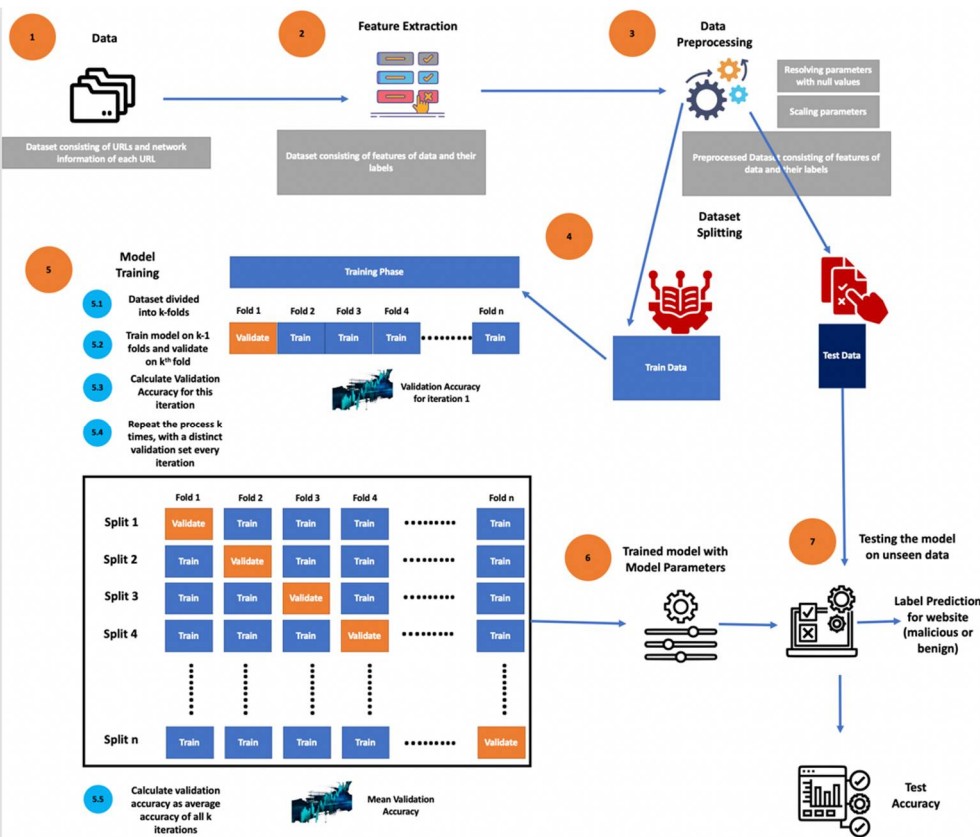

**Figure 9.** The treatment of data and model training and validation in the current study, using 10-fold cross-validation and different data balancing techniques with various classifiers.

*3.6. Evaluation Metrics*

Five metrics have been used in this study to gauge the results and compare different methods. The metrics are calculated using the confusion matrix, where true positives (TPs—number of malicious websites identified correctly as malicious by the classifier), true negatives (TNs—number of benign website records identified correctly as benign by the classifier), false positives (FPs—number of benign websites identified incorrectly as malicious by the classifier), and false negatives (FNs—number of malicious websites incorrectly as benign by the classifier) are used to calculate accuracy, precision, recall, F1-score, and false positive rate (FPR) for each classifier.

3.6.1. Accuracy

Accuracy is the ratio of a correct prediction made by the classifier to the total prediction made by the classifier:

$$Accuracy = \frac{TP + TN}{TP + FP + TN + FN} \tag{1}$$

3.6.2. Precision

In precision, we consider the predictions made by our classifier as our baseline:

$$Precision = \frac{TP}{TP + FP} \tag{2}$$

### 3.6.3. Recall

Recall considers the truth as the baseline and is the ratio of true positives and the total number of positives in the dataset:

$$Recall = \frac{TP}{TP + FN} \tag{3}$$

### 3.6.4. F1-Score

F1-score is the harmonic mean of the precision and recall:

$$F1 - Score = 2 \times \frac{Precision \times Recall}{Precision + Recall} \tag{4}$$

### 3.6.5. False Positive Rate (FPR)

The false positive rate (FPR) is the ratio of false positives, and the total negatives present in the dataset:

$$FPR = \frac{FP}{FP + TN} \tag{5}$$

## 4. Results

This study compares the performance of various classifiers on the "Malicious Websites" dataset to classify which websites are malicious and which are not. The performance of different classifiers was measured using multiple evaluation metrics including accuracy, precision, recall, F1 score, and false positive rate. Since the data was imbalanced, several techniques such as random undersampling, random oversampling, and the SMOTE, were applied to the data for balancing the number of samples for malicious and for benign websites. The results of various classifiers were then computed on balanced and imbalanced data.

### 4.1. Exploratory Data Analysis

The exploratory data analysis on the "Malicious Websites" dataset revealed some interesting insights about the data. Correlation, which describes the relationship between two or more variables, was utilized to determine the relevant features in the feature set that influence the "Type" variable the most; for example, "APP BYTES" or "REMOTE APP" appear to have little influence on the results since they have a low correlation with the target column [38]. On the other hand, "NUMBER SPECIAL" and "URL LENGTH" appear to have a huge influence on the malicious nature of the websites. The correlation plot for each variable with the target variable "Type" is shown in Figure 10.

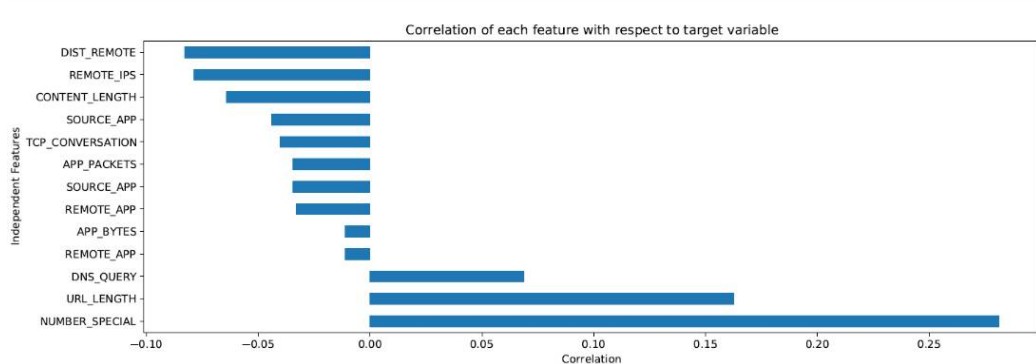

**Figure 10.** A figure displaying the correlation histogram between different input features and the target "Type" variable.

## 4.2. Confusion Matrix

The number of true positives, true negatives, false positives, and false negatives are an important indicator of how good the method is performing in various evaluation metrics. Although the ratio is shown via the accuracy, precision, or recall, the raw numbers are also interesting to view. Table 3 displays the count of TPs, TNs, FPs, and FNs.

**Table 3.** Table displaying the number of true positives (TPs—malicious websites identified as malicious), true negatives (TNs—benign websites identified as benign), false positives (FPs—benign websites identified as malicious), and false negatives (FNs—malicious websites identified as benign by the classifier).

| | **Imbalanced Data** | | | | **SMOTE** | | | |
|---|---|---|---|---|---|---|---|---|
| **Model Name** | **TP** | **FP** | **FN** | **TN** | **TP** | **FP** | **FN** | **TN** |
| Decision Tree | 440 | 20 | **23** | **51** | 396 | 36 | 21 | 486 |
| Random Forest | 454 | 6 | 25 | 49 | 407 | 25 | **7** | **500** |
| SVC | **460** | **0** | 74 | 0 | 281 | 151 | 277 | 230 |
| Logistic Regression | 449 | 0 | 50 | 24 | **454** | **6** | 74 | 0 |
| Stochastic Gradient Decent | 454 | 6 | 74 | 0 | 258 | 174 | 324 | 183 |
| | **Random Under Sampling** | | | | **Random Over Sampling** | | | |
| Model Name | TP | FP | FN | TN | TP | FP | FN | TN |
| Decision Tree | 66 | 5 | 9 | 50 | 420 | 20 | 0 | 499 |
| Random Forest | **67** | **4** | 10 | 49 | **428** | **12** | 0 | 499 |
| SVC | 30 | 41 | 12 | 47 | 122 | 318 | 66 | 433 |
| Logistic Regression | 48 | 23 | 6 | 53 | 287 | 153 | 42 | 457 |
| Stochastic Gradient Decent | 0 | 71 | **0** | **59** | 202 | 238 | 221 | 278 |

## 4.3. Accuracy

Among all the classifiers, random forest shows the highest accuracy, which is 0.97 when data was balanced after applying the oversampling technique. The accuracy scores of different classifiers along with different methods for making the data to be balanced as well as the imbalanced data are shown in Figure 11.

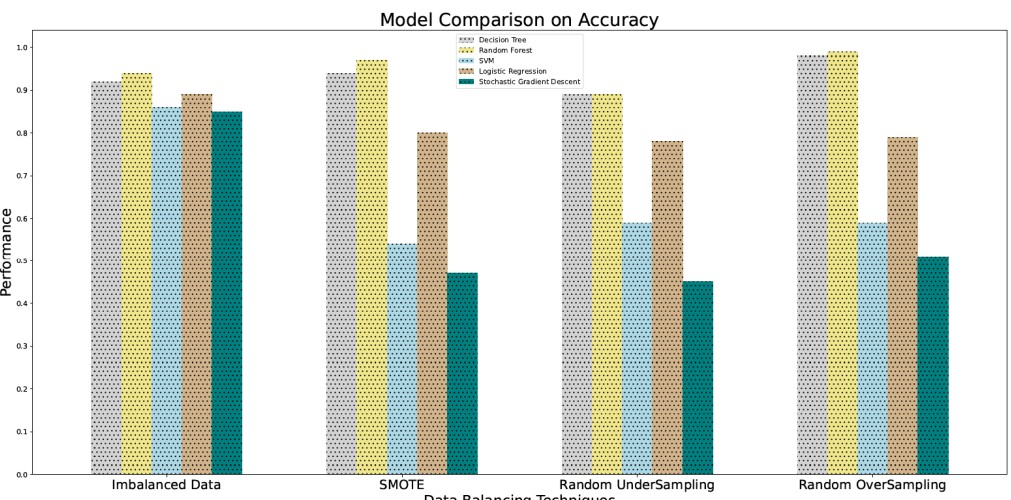

**Figure 11.** An illustration of performance comparison of different machine learning models on various data balancing techniques using accuracy as the evaluation metric.

### 4.4. Precision

The highest precision of 0.99 can be achieved by random forest when the oversampling technique for making the data balance is applied. The results of another classifier on different methods including imbalanced data, and data balancing techniques including undersampling, oversampling, and the SMOTE are shown in Figure 12.

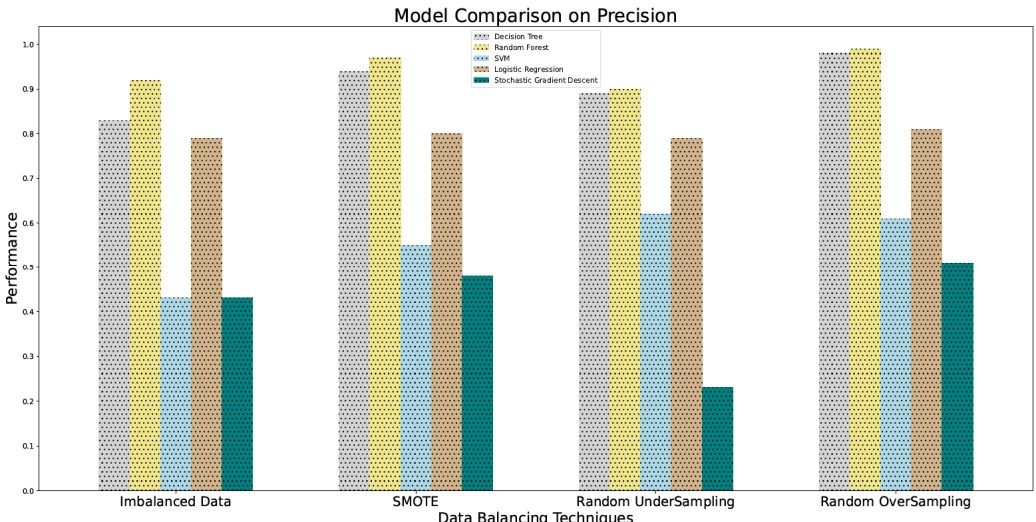

**Figure 12.** An illustration of performance comparison of different machine learning models on various data balancing techniques using precision as the evaluation metric.

### 4.5. Recall

Among all of the models, random forest and decision tree give better recall, which is 1 when data was balanced after applying the oversampling. The result of different classifiers on imbalanced data, undersampling, oversampling, and the SMOTE are shown in Figure 13.

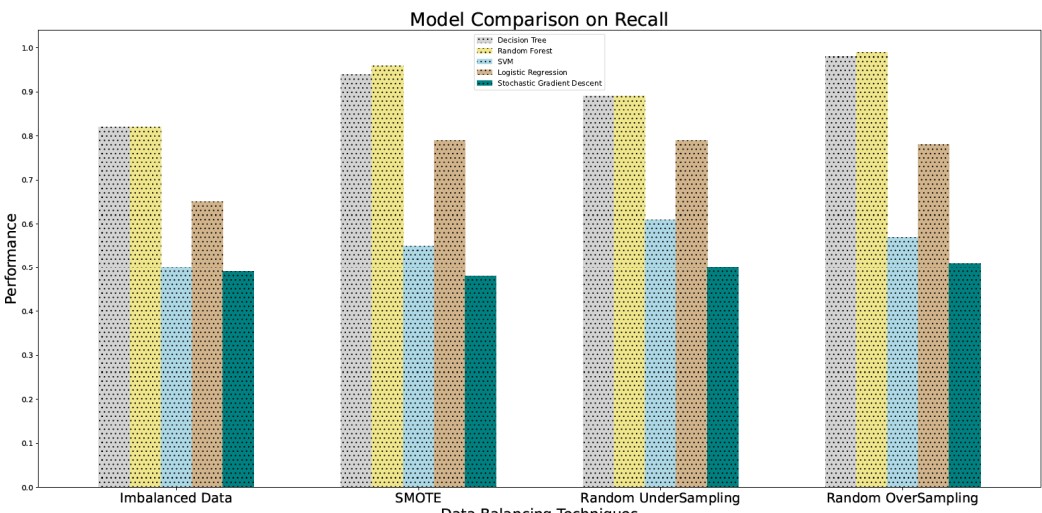

**Figure 13.** An illustration of performance comparison of different machine learning models on various data balancing techniques using recall as the evaluation metric.

### 4.6. F1-Score

The highest F1-score can be achieved from random forest when trained on balanced data using the oversampling technique. While the F1-score results of other classifiers along with imbalanced data, and balanced data using undersampling, oversampling, and the SMOTE are mentioned in Figure 14.

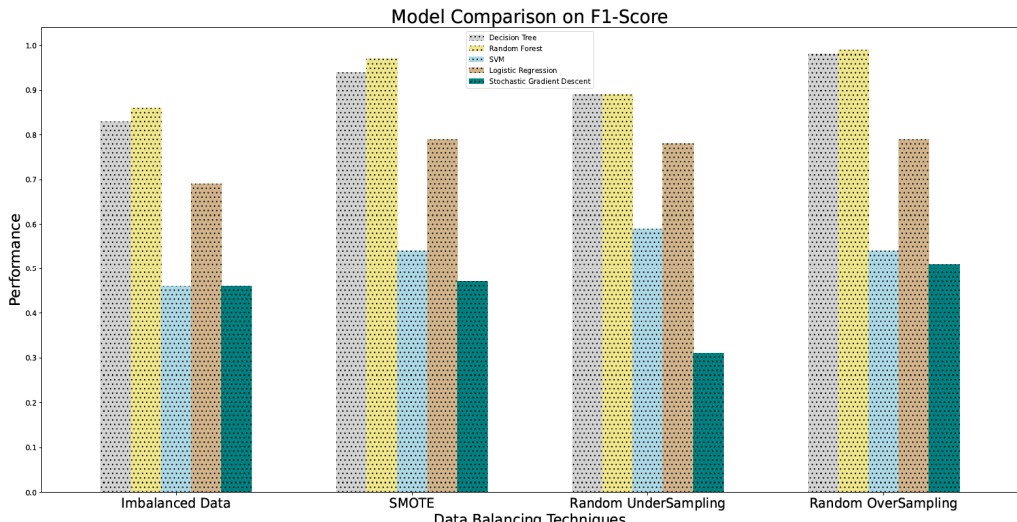

**Figure 14.** An illustration of performance comparison of different machine learning models on various data balancing techniques using F1-score as the evaluation metric.

*4.7. False Positive Rate (FPR)*

The FPR is the likelihood of a false alarm being raised: that a positive result will be returned when the true value is negative. So, a lower value of the FPR is always preferred. The logistic regression gives the lowest false positive rate of 0.01 when trained on the imbalanced dataset. However, random forest and stochastic gradient descent give a second lowest false positive rate of 0.05. However, random forest also gives the lowest false error rate of 0.05, when trained on the balanced dataset using the oversampling technique. Figure 15 shows the performance of various classifiers based on the false positive rate (FPR).

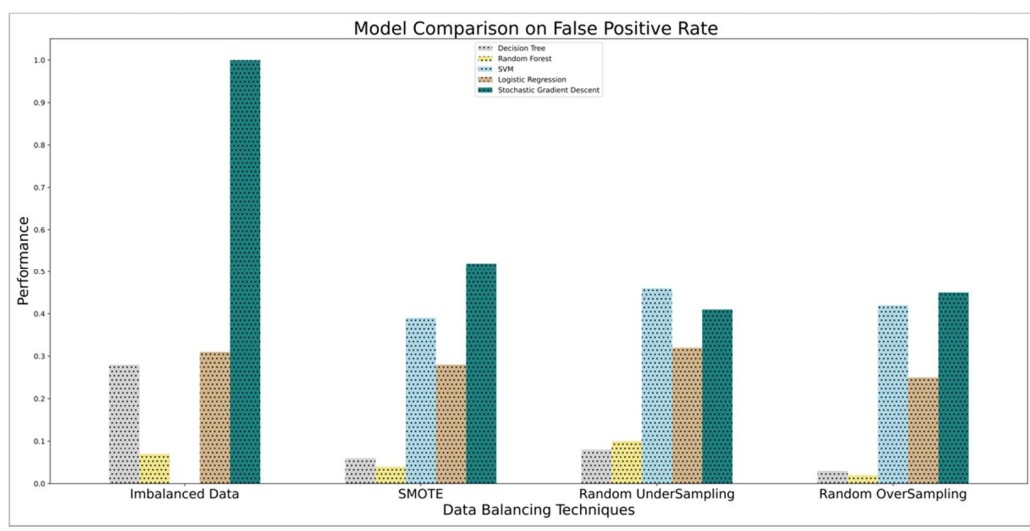

**Figure 15.** An illustration of performance comparison of different machine learning models on various data balancing techniques using the false positive rate as the evaluation metric.

## 5. Discussion

This study tested multiple classifiers with a 70/30 training split and 10-fold cross validation for validating the trained models. We first conducted exploratory data analysis to estimate the best features and parameter settings for the machine learning classifiers used in this study. This ensured that the most optimized set of parameters were deployed, and the performance of the current methodology was optimized for the given data set. Multiple machine learning classifiers were also used in this study.

## 5.1. Using 10-Fold Cross Validation

Using a 10-fold cross-validation approach ensured that the training data and validation data were continually mixed and re-used in training. Hence, the validation took place on the whole data. The final training accuracy is the average accuracy across all validation samples. However, a possible drawback is that the data used for validation is not completely unseen and may not agree with the results when checked on completely unseen data, for example, with the test data. Hence, typically separate test data is always required in the case of the k-fold cross-validation method.

## 5.2. Using Multiple Data Balancing Techniques

We also tested multiple data balancing techniques to assure that the model does not learn any bias towards a certain class. It also helped us verify the results of various data balancing techniques for the malicious and benign website dataset. The data balancing techniques play an essential role in identifying and removing bias, as otherwise a class that has the greatest number of samples, may dominate the model training. If the model learns best to classify just the majority class, it will by default perform better than a random method, but does not have the ability to work for any other class. This means that the results and performance will be heavily biased towards the majority class only. Hence, to ensure a fair judgement of method performance, data balancing is necessary.

## 5.3. Evaluation with Multiple Evaluation Metrics

Using multiple evaluation metrics also ensures that the models were evaluated based not only on performance in a specific criterion, but on the overall performance across categories. It also reflected if a particular measure or a particular class affected the results. For example, the accuracy is a measure that consists of ratio of correctly identified malicious and non-malicious websites with the total number of websites. While accuracy alone is an excellent measure of correct classification, note that if the number of malicious websites in the dataset are too few, as compared to the number of benign websites, the accuracy will be dependent on the performance in the negatives mainly. In such a case, a false positive rate (FPR) is a true reflection of system performance. Therefore, we showed performance of each classifier in all five parameters. Each evaluator indicates how the model performs w.r.t. different performance criteria. As is apparent from Figures 11–15, random forest with the SMOTE data balancing technique gives the best results across all five indicators.

## 5.4. Overall Prospects

Using a completely isolated test dataset, as well as utilizing k-fold cross-validation and data balancing techniques, ensured that overfitting was avoided, and minimum bias towards a specific class was introduced. The method achieved good training and testing accuracy, where the testing accuracy was slightly lower than the training accuracy as expected, since the test dataset is completely unseen for the model while it is somewhat seen for the validation during the training phase.

## 6. Conclusions

Digital security is one of the paramount concerns in today's digital world, where billions of dollars are lost to digital theft every year. In that aspect, malicious websites are the most common source of digital theft and accurate differentiation between malicious and benign websites is desired. We have used several machine learning algorithms in this study and measured the performance of each algorithm using several evaluation metrics such as F1-score, precision, recall, accuracy, and false positive rate. The machine learning algorithms demonstrated in this work included decision trees, random forest, the SVMs, logistic regression, and stochastic gradient descent. The "Malicious Website" data used in this study is imbalanced and heavily biased toward malicious websites. Therefore, several data balancing techniques were also evaluated in this study to measure their effectiveness. A 10-fold cross-validation technique was used during the training phase to remove any

effects of poor sampling. Out of all the machine learning algorithms studied in this work, and with all the data balancing techniques, random forest showed the best results when trained on the dataset after balancing it using the random oversampling technique. It was closely followed by random forest with a balanced dataset using the SMOTE technique. In short, the study demonstrates that the performance of machine learning algorithms is vastly influenced by the data and its properties, and random forest offers a significant performance advantage when used with a balanced dataset.

**Author Contributions:** Conceptualization, R.H.A.; formal analysis, I.U.H.; investigation, T.A.K.; methodology, R.K.; software, Z.U.A.; writing—original draft preparation, I.U.H.; writing—review and editing, R.H.A. and T.A.K.; visualization, Z.U.A.; supervision, R.K.; project administration, R.K. All authors have read and agreed to the published version of the manuscript.

**Funding:** This research received no external funding.

**Institutional Review Board Statement:** Not applicable.

**Informed Consent Statement:** Not applicable.

**Data Availability Statement:** The data is publicly available and can be found at the Kaggle website https://www.kaggle.com/code/angadchau/maliciousurldetection/data (accessed on 31 August 2022).

**Conflicts of Interest:** The authors declare no conflict of interest.

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
