# Peer review of "Significance of Machine Learning for Detection of Malicious Websites on an Unbalanced Dataset"

_digital, doi:10.3390/digital2040027_

Round 1

Reviewer 1 Report

The paper describes the application of machine learning techniques on an unbalanced dataset, and applies the techniques on a malicious website dataset. Overall the paper is interesting and should be ok for publication with a few minor corrections related to the written text.

The main things noted are for rewriting some parts of the text, which were either unclear or needed revision. Please make the following suggested changes to improve the quality of this manuscript.

Line # 53 - 56 on Page 2 (2nd para Introduction) - An example or two of such incidents would be appreciated.

Lines # 79 - 85 on Page 3 - A table describing the confusion matrix instead of textual description would be better. Plus, short explanation of what TP, TN, DP and FN are would also be good.

Lines # 117 - 120 on Page 3 - Both the lines seem to be repeated. Kindly write one sentence only.

Lines 184 on Page 5 - Please explain or rewrite the sentence "lot of data".

Lines 190 - 192 on Page 5 - Performance is significantly lower on what evaluation criteria?

Lines 194 - 195 on page 5 - The table appears problematic/confusing given comparison is difficult between various approaches as evaluation criteria differs. Either separate the metrics into different columns or write one metric only.

Table 1 - Metric should be precision, not precious.

Line 211 on page 7 -  Rewrite needed.

Line 271 on page 8 - better to be more consistent with the use of capitalizations.

Line 281 - 295 on page 9 - Logistic regression explanation requires rewrite.

Line 324 on page 11 - Please explain more "the process may take eons".

Line 340-341 on page 11 - "By way of illustration" seems non-scientific. Please rewrite.

Author Response

As suggested, all revisions have been made. The major part was rewriting, which has been incorporated and text cooler is changed to red to indicate changes.

Line # 53 - 56 on Page 2 (2nd para Introduction) - An example or two of such incidents would be appreciated.

Table 1 has been added that gives recent examples of such incidents.

Lines # 79 - 85 on Page 3 - A table describing the confusion matrix instead of textual description would be better. Plus, short explanation of what TP, TN, DP and FN are would also be good.

As suggested, a table displaying the confusion matrix is now added to Results. Short explanation of True Positives (TPs), True Negatives (TNs), False Positives (FPs), and False Negatives (FNs) has also been added.

Lines # 117 - 120 on Page 3 - Both the lines seem to be repeated. Kindly write one sentence only.

Removed one line. Thanks for pointing out.

Lines 184 on Page 5 - Please explain or rewrite the sentence "lot of data".

The line has been rewritten. Thanks.

Lines 190 - 192 on Page 5 - Performance is significantly lower on what evaluation criteria?

Thanks for pointing out. The performance criteria has been mentioned in the line.

Lines 194 - 195 on page 5 - The table appears problematic/confusing given comparison is difficult between various approaches as evaluation criteria differs. Either separate the metrics into different columns or write one metric only.

Agreed with the reviewer.  Only the values of accuracy are now mentioned in the table.

Table 1 - Metric should be precision, not precious.

Thanks for pointing out. The mistake has been corrected.

Line 211 on page 7 -  Rewrite needed.

The line has been rewritten.

Line 271 on page 8 - better to be more consistent with the use of capitalizations.

Agreed. All classifiers now start with lower case letters consistently.

Line 281 - 295 on page 9 - Logistic regression explanation requires rewrite.

Agreed. The whole paragraph has been rewritten.

Line 324 on page 11 - Please explain more "the process may take eons".

Agreed. The sentence is rewritten to make the explanation more explicit and crisp.

Line 340-341 on page 11 - "By way of illustration" seems non-scientific. Please rewrite.

Agreed. The sentence has been rewritten. Thanks.

Reviewer 2 Report

1) In introduction, authors are suggested to introduce the problem, motivate the problem, and summarize the main contributions in detail

2) Some related works can be included and they are 

https://content.iospress.com/articles/journal-of-intelligent-and-fuzzy-systems/ifs169429

https://ieeexplore.ieee.org/abstract/document/8494159

3) Compare the performance of the proposed method with the exsisting 3 methods

4) Discuss the advatages and limitations of the proposed method

5) How the dataset is divided into training and testing? Is testing dataset is completley unseen from training dataset. Is there any validation dataset is used in experiments? If not, how the parameters of machine learning algorithm are identifed. Are these parameters are optimal

Author Response

The reviewer has raised some very interesting points for the revision of the manuscript and has suggested some excellent points. We have tried to address their concerns and incorporate their suggestions to the best of our knowledge.

1) In introduction, authors are suggested to introduce the problem, motivate the problem, and summarize the main contributions in detail

Agreed. We have tried to revise our introduction and make it more crisp, and direct. We now introduce the problem in the first paragraph, and quickly build towards the solution. The problem is motivated in the next two paragraphs and is followed by our contributions in the final paragraph of the introduction.

2) Some related works can be included and they are 

https://content.iospress.com/articles/journal-of-intelligent-and-fuzzy-systems/ifs169429

https://ieeexplore.ieee.org/abstract/document/8494159

Agreed. Both works have been added to Table 2 as part of Literature Review.

3) Compare the performance of the proposed method with the exsisting 3 methods

The study already compares five different classifiers, and those classifiers are shown to work on similar datasets in other studies.

4) Discuss the advatages and limitations of the proposed method

A separate Discussion section has now been added to address the advantages and disadvantages of the proposed method.

5) How the dataset is divided into training and testing? Is testing dataset is completley unseen from training dataset. Is there any validation dataset is used in experiments? If not, how the parameters of machine learning algorithm are identifed. Are these parameters are optimal

Thanks for pointing out. The division of dataset into training, validation, and testing data has now been done and described in further detail with a separate paragraph in Methodology. The description for k-fold cross-validation approach has also been updated. The testing data (30%) is completely unseen for the model. The validation dataset is the same as the training dataset, except that for each iteration it is the tenth part on rotational basis. The parameters have been optimised by using a variety of settings, and then choosing the one producing the highest accuracy for the given model.